# The Non-Fibrillating N-Terminal of α-Synuclein Binds and Co-Fibrillates with Heparin

**DOI:** 10.3390/biom10081192

**Published:** 2020-08-16

**Authors:** Line K. Skaanning, Angelo Santoro, Thomas Skamris, Jacob Hertz Martinsen, Anna Maria D’Ursi, Saskia Bucciarelli, Bente Vestergaard, Katrine Bugge, Annette Eva Langkilde, Birthe B. Kragelund

**Affiliations:** 1Department of Drug Design and Pharmacology, University of Copenhagen, Universitetsparken 2, 2100 Copenhagen, Denmark; lkskaanning@gmail.com (L.K.S.); thomas.s.pedersen@sund.ku.dk (T.S.); jacob.martinsen@bio.ku.dk (J.H.M.); saskia.bucciarelli@gmail.com (S.B.); bente.vestergaard@sund.ku.dk (B.V.); 2Structural Biology and NMR Laboratory, Department of Biology, University of Copenhagen, Ole Maaløes vej 5, 2200 Copenhagen N, Denmark; asantoro@unisa.it (A.S.); katrine.bugge@bio.ku.dk (K.B.); 3Department of Pharmacy, University of Salerno, Via Giovanni Paolo II, 132, 84084 Fisciano, Italy; dursi@unisa.it

**Keywords:** α-synuclein, heparin, fibrillation, NMR, binding, IDP, intrinsically disordered proteins, SAXS, Parkinson’s disease, type I β-turn

## Abstract

The intrinsically disordered protein α-synuclein (aSN) is, in its fibrillated state, the main component of Lewy bodies—hallmarks of Parkinson’s disease. Additional Lewy body components include glycosaminoglycans, including heparan sulfate proteoglycans. In humans, heparan sulfate has, in an age-dependent manner, shown increased levels of sulfation. Heparin, a highly sulfated glycosaminoglycan, is a relevant mimic for mature heparan sulfate and has been shown to influence aSN fibrillation. Here, we decompose the underlying properties of the interaction between heparin and aSN and the effect of heparin on fibrillation. Via the isolation of the first 61 residues of aSN, which lacked intrinsic fibrillation propensity, fibrillation could be induced by heparin, and access to the initial steps in fibrillation was possible. Here, structural changes with shifts from disorder via type I β-turns to β-sheets were revealed, correlating with an increase in the aSN_1–61_/heparin molar ratio. Fluorescence microscopy revealed that heparin and aSN_1–61_ co-exist in the final fibrils. We conclude that heparin can induce the fibrillation of aSN_1–61_, through binding to the N-terminal with an affinity that is higher in the truncated form of aSN. It does so by specifically modulating the structure of aSN via the formation of type I β-turn structures likely critical for triggering aSN fibrillation.

## 1. Introduction

The intrinsically disordered protein (IDP) α-synuclein (aSN) [1] locates predominantly in the presynaptic termini in neurons, with a cytosolic concentration of 30–60 μM [2]. The physiological function of aSN is still highly debated, but different roles have been suggested, including regulatory effects on vesicular transport such as exo- and endocytosis [3], dopamine transmission [4], or protecting the cell from oxidative stress [5].

The 140 residues of aSN can be divided into three regions: the N-terminal region (residues 1–60), the non-amyloid-β component (NAC) region (residues 61–95) and the C-terminal region (residues 96–140) [6]. The N-terminal region has been shown to play an important role in interactions with membranes and vesicles, both in vitro [7] and in vivo [8]. In this region and the NAC region, aSN has a total of six evolutionary conserved K[T/A]K[E/Q][G/Q]V-motifs, which are believed to be important for its membrane interaction [6].

aSN has been linked to different neurodegenerative diseases referred to as synucleinopathies. In Parkinson’s disease (PD), cellular deposits found in the substantia nigra, called Lewy bodies, are rich in aSN [9]. Here, aSN is found in fibrillar forms with structural characteristics of amyloid fibrils [10,11]. The cross-β core of aSN fibrils is formed by the NAC region, but recent high-resolution structures obtained by cryo-electron microscopy (EM) have shown that also part of the N-terminal partakes in the fibril β-structure [12]. Supported in part by solid-state nuclear magnetic resonance (NMR) spectroscopy, the fibril core structure was shown to include residues 38–95, revealing flexible structures for the N-terminal until residue 22 and for the C-terminal onwards from residue 105 [13]. The pathological role of the N-terminal is not clear, but it is remarkable that all known single-point mutations related to PD locate to the N-terminal region (A18–A53) and not the NAC region [14]. While previous studies suggested the N-terminal as inhibitory for fibrillation [15], recent studies have shown that N-terminal fragments, e.g., 15–65, can self-aggregate and, although the structural nature of these aggregates was not fully determined, accelerate aggregation of full-length aSN [16].

Heparan sulfate proteoglycans (HSPGs) have recently been shown to be present in Lewy bodies from PD patients [17]. Proteoglycans are heavily glycosylated proteins with covalently attached glycosaminoglycans (GAGs), which are linear unbranched polysaccharides with a heterogeneous structure [18,19], playing diverse physiological roles in cell–cell interaction, development, migration and growth [20,21]. In addition, fibrillar and aggregate uptake in the cell may occur via binding to HSPGs on the cell surface or selected glycosylated cell surface proteins [22,23,24]; thus, GAGs may play a significant role in the spreading and further seeding of intracellular aggregation. In humans, heparan sulfate has been shown to increase its sulfation level with age, leading to an age-dependent decoration of sulfates [25]. One of the most sulfated GAGs is heparin [18], which is structurally closely related to heparan sulfate. Thus, heparin is a highly relevant mimic of mature heparin sulfate. Heparin is known to interact with proteins through Cardin–Weintraub sequences, where basic residues are placed either in a linear pattern with hydropathic residues [26] or via a folded structure bringing sequentially distant residues close [20]. The cationic (C) and polar (P) CPC clip motif is another heparin-binding motif, where two cationic residues separated by a polar residue facilitate heparin binding [27]. It has been reported that heparin influences the fibrillation of aSN in a concentration-dependent manner either through ionic interaction with the N-terminal [28,29] or via the stabilization of a transition state favoring the β-sheet structure [29,30]. Furthermore, heparin has been found integrated in fibrils [29]. Although the structural details are not known, heparin-induced fibrils of aSN have a different morphology to aSN fibrillated without heparin [28,31]. In addition, heparin may induce the fibrillation of proteins that are not inherently found to fibrillate [32,33]. Finally, GAGs have also been located in the amyloidogenic deposits of other neurodegenerative diseases [19], suggesting a potential link between the presence of GAGs and fibrillation. However, to date, it has not been addressed which properties of GAGs—and, in particular, heparin—can modulate fibrillation.

In this study, we addressed the interplay between heparin and aSN. To remove complications from fibrillation, we focused on the aSN N-terminal (aSN_1–61_) in isolation and showed it to be as disordered as full-length aSN. We show that heparin binds directly to the N-terminal end of aSN_1–61_ and does so with a higher affinity than to full-length aSN. While we did not observe fibrillation of aSN_1–61_ on its own, we intriguingly also observed heparin-induced fibrillation of aSN_1–61_ and detected heparin as a co-existing partner of the fibrils. The co-existing fibrils of aSN_1–61_/heparin accelerated the fibrillation of full-length aSN as well as the further fibrillation of aSN_1–61_. In addition, our data suggest that no individual property of heparin but rather the combined physical and chemical characteristics modulate the structure of aSN by inducing type I β-turn structures, which may be critical for triggering aSN fibrillation.

## 2. Materials and Methods

### 2.1. Compounds

Heparin was purchased from Iduron (HEP001). It was dissolved in 50 mM sodium phosphate, pH 7.4, with 0–500 mM NaCl and stored at 4 °C for a maximum of two days before use. Chondroitin sulfate A from bovine trachea and β-casein from bovine milk were purchased from Sigma-Aldrich (C9819 and C6905, Søborg, Denmark).

### 2.2. Expression and Purification of Recombinant aSN_1–61_

A pET-11a vector with an insert for His-SUMO-aSN_1–61_ was purchased from GenScript^®^ (Leiden, Netherlands). BL21 (DE3) *E. coli* cells were heat-shock transformed with the plasmid, incubated in Super Optimal Broth medium for 1 h and plated on agar containing 1:1000 ampicillin (100 mg/mL). The plates were incubated overnight at 37 °C, and preheated Luria Bertani (LB) medium (50 mL) with 1:1000 ampicillin (100 mg/mL) was inoculated with one colony and incubated overnight at 37 °C and 200 rpm. Glycerol stocks were made from the overnight culture by mixing them 1:1 (volume) with 86% (v/v) sterile glycerol, aliquoted in 0.5 mL and stored at −80 °C. For expression, an aliquot of 2 μL of the glycerol stock, or a colony from the agar plate, was transferred to preheated LB medium with 1:1000 of ampicillin (100 mg/mL) and incubated overnight at 37 °C and 200 rpm. The LB expression cultures, each 1 L, contained 1:1000 of ampicillin (100 mg/mL) and were inoculated with 1:100 of the overnight culture and incubated at 37 °C and 200 rpm. The expression of His-SUMO-aSN_1–61_ was induced with 1:1000 of IPTG (1 M) at OD_600_ = 0.5–0.6. The cells were harvested 2 h after induction by centrifugation at 4801× *g* for 20 min at 4 °C. The cells were dissolved in 30 mL of buffer A (50 mM Tris, 150 mM NaCl, 10 mM Imidazole, pH 8) per L of expression medium. The cell pellet was stored at −20 °C until purification.

The resuspended cell pellet was lysed by six cycles of thawing and freezing in liquid nitrogen. The lysed cells were centrifuged at 25,515× *g* for 20 min at 4 °C. The supernatant was loaded onto a 5 mL HisTrap FF column (GE healthcare, Chicago, IL, USA) equilibrated in buffer A with a flow of 1 mL/min. His-SUMO-aSN_1–61_ was eluted from the column using a step gradient with imidazole—10, 200, and 500 mM imidazole—with at flow of 2 mL/min. Fractions of 5 mL were collected and analyzed by SDS-PAGE, fractions containing His-SUMO-aSN_1–61_ were pooled, and 0.5–1 mL of 10 μM His-ULP1 was added (see below for production details). The solution with His-SUMO-aSN_1–61_ and His-ULP1 was subsequently dialyzed overnight at 4 °C in 50 mM Tris, 150 mM NaCl, 10 mM imidazole, 1 mM 2-mercaptoethanol, pH 8, in dialysis bags with a cut-off of 3.5 kDa. The dialysate was loaded onto a HisTrap FF column equilibrated in buffer A. The flow-through containing the cleaved aSN_1–61_ was collected in fractions of 5 mL. aSN_1–61_ was dialyzed into 50 mM Tris, 10 mM NaCl, pH 8.5, using a cut-off of 3.5 kDa. The dialysate was loaded onto an HiTrap FF SP 5 mL column (GE healthcare), equilibrated with 50 mM Tris, 10 mM NaCl, pH 8.5, and aSN_1–61_ was eluted with a 0–100% step gradient to 0.5 M NaCl using a flow rate of 1 mL/min. aSN_1–61_ was collected in fractions of 0.5 mL. Afterwards were the fractions with aSN_1–61_ dialyzed overnight at 4 °C in a 50 mM sodium phosphate buffer, 150 mM NaCl, pH 7.4, with a Spectra-Por^®^ Float-A-Lyzer^®^ G2 (Merck, Darmstadt, Germany), with a cutoff of 3.5 kDa.

The concentration of aSN_1–61_ was determined using a Pierce™ BCA Protein Assay Kit (ThermoFisher 23225, Denver, CO, USA), following the standard protocol for microplate mode. Incubation and measurements were performed using a FLUOstar OMEGA plate reader (BMG LABTECH, Orthenberg, Germany). The identity of the purified aSN_1–61_ was confirmed by amino acid analysis by ALPHALYSE™ (Odense, Denmark).

### 2.3. Purification of Full-Length Recombinant aSN

The purification of full-length aSN was essentially performed as previously described [34] with a few modifications. Briefly, aSN was expressed in LB medium with 1:1000 ampicillin (100 mg/mL), and protein expression was induced at OD_600_ = 0.6. The cells were harvested after 3 h at 4801× *g*, and the cell pellet was resuspended and stored at −20 °C. The cells were lysed with osmotic shock, and non-thermostable proteins were removed by boiling and centrifugation. The supernatant was dialyzed overnight in 20 mM Na_2_HPO_4_, 2.5 mM EDTA at pH 6.5. The dialyzed solution was filtered through a 0.45 μm filter and loaded onto a 5 mL HiTrap Q FF column (GE healthcare, Chicago, IL, USA) equilibrated in 20 mM Na_2_HPO_4_, 2.5 mM EDTA at pH 6.5. aSN was eluted by a linear gradient with a final salt concentration of 0.5 M NaCl. Fractions containing aSN were pooled and loaded onto a Superdex 200 size exclusion column 120:100 (GE Healthcare, Chicago, USA), which was equilibrated in 50 mM Na_2_HPO_4_, 150 mM NaCl, pH 7.4. Fractions containing aSN were pooled at stored at −80 °C. Before the experiments, aSN was dialyzed with a cut-off of 3.5 kDa into 50 mM sodium phosphate, 150 mM NaCl, pH 7.4.

### 2.4. Expression and Purification of Recombinant His-ULP1

His_6_-tagged ULP1 was purified as described in Prestel et al., 2019 [35].

### 2.5. Protein Fibrillation Assay

Thioflavin T (ThT) detected fibrillation assays were conducted using a FLUOstar OMEGA plate reader (BMG LABTECH, Orthenberg, Germany). Bottom–bottom mode was applied with excitation at 450 ± 5 nm, and emission at 480 ± 5 nm was measured. The concentration of aSN_1–61_ was 100 or 200 μM, and for aSN, a concentration of 83 μM (1.2 mg/mL) was used. Varying molar ratios of heparin were included as denoted in the results. In general, all assays were performed in 50 mM NaP, 150 mM NaCl, while a few additional tests were performed in 1 M NaCl, 160 mM (NH_4_)_2_SO_4_ or in the presence of β-casein to decipher the influence of heparin.

Samples including 20 μM ThT were placed in NUNC 96-well optical polymer-based black plates (ThermoFisher 265301, Denver, USA) with a total volume of 150 μL per well and sealed with clear polyolefin tape (ThermoFisher 232702, Denver, CO, USA). The fluorescence was measured every 6 min, in repeated cycles, with orbital shaking (700 rpm) for 280 s of each 360 s cycle. The assays were run at 37 °C for a total of 7 days.

### 2.6. Seeding Experiments

Seed fibrils of aSN_1–61_/heparin was prepared as described above (aSN_1–61_ 200 μM, heparin 333 μM). Subsequently, fibrillation assays for full-length aSN (60 μM) and aSN_1–61_ (200 μM) were performed under four different conditions: (i) in the presence of 10% (v/v) seed fibrils; (ii) in the presence of heparin (33 μM); (iii) in the presence of 10% (v/v) seed fibrils and additional free heparin (33 μM), i.e., a mix of conditions (i) and (ii); and lastly, as control, (iv) aSN and aSN_1–61_ alone, respectively.

### 2.7. Negative Stain Transmission Electron Microscopy

Samples were diluted with MQ water to a protein concentration of 15 μM and spotted onto the carbon-coated Cu grid (FCF, carbon film, 400 MESH Copper, AGAR Scientific, Essex, UK) using one of two different staining procedures. Initially, 10 μL of sample was incubated on the grid for 60 s, the sample solution was removed with filter paper, and 10 μL of 2% (w/v) uranyl acetate was added. After 60 s, 10 μL of MQ water was added to the uranyl acetate and the solution was removed. Finally, the grid was washed with an additional 10 μL of MQ water. In the second protocol, 3 μL of sample was added to the grid and incubated for 60 s before the solution was blotted off and 3 μL of 2% (w/v) uranyl acetate was added and removed after 60 s. Micrographs were collected with a Morgagni 268 microscope (FEI, Hillsboro, OR, USA) in the Core Facility for Integrated Microscopy (CFIM) at the University of Copenhagen, Denmark.

### 2.8. Circular Dichroism (CD)

A Jasco J-1500 CD spectropolarimeter (Jasco, Easton, PA, USA) was used with the following settings: spectral width, 260–190 nm; data pitch, 0.1 mm; scanning speed, 50 nm/min; band width, 1 mm; temperature, 25 °C. A quartz cuvette of 0.1 mm was used, and each spectrum was an accumulation of three scans. Samples before and after the fibrillation assays were diluted to concentrations in the range of 15–20 μM. Background spectra of the buffers (50 mM sodium phosphate (NaP), 150 mM NaCl, pH 7.4, or 50 mM NaP, 1 M NaCl, pH 7.4) were collected at identical settings and subtracted.

### 2.9. Small Angle X-ray Scattering (SAXS)

SAXS data were collected at the beamline P12 [36] operated by EMBL at the Petra III storage ring (DESY, Hamburg, Germany). Data were collected for aSN_1–61_ (223–573 μM) in 50 mM NaP, 150 mM NaCl, pH 7.4 at 8 °C, and for heparin (33–333 μM) in several different buffer conditions: 50 mM NaP with 0–500 mM NaCl pH 7.4, or 50 mM Tris with 0–500 mM NaCl, pH 7.4, at room temperature. The automated sample changer was used with a sample loading volume of 30 μL and sample flow and exposure time of 0.495 s/frame, with 40 frames collected per sample. The scattering profiles of matching buffers were measured before and after each sample. Initial data reduction was performed using the SASFLOW pipeline [37], and primary data analysis was subsequently conducted in PRIMUS [38]. The ensemble optimization method (EOM) [39] was employed to model the conformational ensemble of aSN_1–61_. A fit to the experimental data was obtained through selection from a random pool of 10,000 structures generated based on the sequence, using default parameters with native-like chains.

### 2.10. NMR Sample Preparation

For NMR analyses, full-length aSN or aSN_1–61_ was expressed in M9 minimal medium with the addition of ^15^NH_4_Cl (and ^13^C6-glucose when needed) and purified as described previously [40] and detailed above. After dialysis into 20 mM Na_2_HPO_4_, 150 mM NaCl, pH 7.4, the samples were added to 10% (v/v) D_2_O, 0.02% (w/v) sodium azide (NaN_3_), and 0.7 mM 4,4-dimethyl-4-silapentane-1-sulfonic acid (DSS). For the acquisition of spectra for the backbone assignment of ^13^C,^15^N-aSN_1–61_, the protein concentration was 0.5 mM. For measurements of the hydrodynamic radii (*R*_h_) of ^13^C-^15^N-aSN_1–61_ and full-length ^15^N-aSN, a sample concentration of 0.1 mM protein with 0.2% 1,4-dioxane added as an internal reference was used. The interaction of ^13^C,^15^N-aSN_1–61_ with heparin was evaluated by monitoring the changes in chemical shifts using solutions of 0.1 mM ^13^C,^15^N-aSN_1–61_ prepared in duplicate. To one solution, 1 mM heparin was added. Using the two solutions, different molar ratios of ^13^C,^15^N-aSN_1–61_/heparin (1:0, 1:0.3, 1:0.5, 1:1, 1:2, 1:4, 1:6, 1:8 and 1:10) were obtained. Similar titration was performed with ^15^N-aSN using aSN/heparin ratios of (1:0, 1:0.5, 1:1, 1:2, 1:4, 1:6, 1:8 and 1:10) and an aSN concentration of 0.1 mM.

### 2.11. NMR Data Recording and Processing

All NMR spectra were acquired at 10 °C (283 K) on a Bruker AVANCE III 600MHz (^1^H) spectrometer equipped with a cryogenic probe. The spectra were transformed and visualized in Topspin (Bruker, Fälladen, Switzerland) and analyzed using the CcpNMR Analysis software [41]. Assignments of the backbone nuclei of ^13^C,^15^N-aSN_1–61_ were performed manually from the analysis of ^1^H,^15^N-HSQC, HNCACB, HNCOCACB, HN(CA)CO, HNCO and HN(CA)NNH spectra acquired with non-uniform sampling [42] using standard pulse sequences. Proton chemical shifts were referenced internally to DSS at 0.00 ppm, with heteronuclei referenced by relative gyromagnetic ratios.

Diffusion constants were acquired by diffusion ordered spectroscopy (DOSY) experiments [43]. A total of 32 spectra with gradient strengths ranging from 2% to 98% of the maximum value were recorded. A diffusion time Δ of 200 ms and gradient length *δ* of 2 ms were used in all the experiments. The pseudo 2D data were Fourier transformed, baseline corrected in Topspin and subsequently analyzed in Dynamics Center (Bruker, Fälladen, Switzerland). The diffusion constants were determined by the fitting of peak intensity decays using the Stejskal–Tanner equation [44]:(1)$$I=I_{0}e^{-g^{2}\gamma^{2}\delta^{2}\left(\Delta -\frac{\delta }{3}\right)D}$$
where *I* is the intensity, *g* is the gradient strength, *γ* is the gyromagnetic ratio of ^1^H, and *D* is the diffusion constant. 1,4-dioxane with a known hydrodynamic radius of 2.12 Å was used as an internal reference and used for the calculation of *R_h_* according to Wilkins et al. [45]:(2)$$R_{h}^{prot}=R_{h}^{ref}\cdot \frac{D_{t}^{ref}}{D_{t}^{prot}}$$
where *D^ref^* and $R_{h}^{ref}$ are the diffusion and the hydrodynamic radius of the internal reference, respectively, and *D^prot^* and $R_{h}^{prot}$, the diffusion and the hydrodynamic radius of the protein.

For chemical shift perturbation (CSP) analysis, ^1^H-^15^N-HSQC spectra of aSN_1–61_ free and with increasing concentrations of heparin were recorded and analyzed. Chemical shift perturbations (CSPs) were analyzed as combined amide chemical shift changes following the expression:(3)$$\Delta \delta_{NH}\left(ppm\right)=\sqrt{{\left(\Delta \delta {\;}^{1}H\right)}^{2}+{\left(0.154\;\Delta \delta {\;}^{15}N\right)}^{2}}$$

The determination of the binding affinity, *K_D_*, from changes in chemical shift was performed as described [46] using the series of ^1^H,^15^N-HSQC spectra recorded for free aSN_1–61_ and with increasing heparin concentration. The content of the transient structure in aSN_1–61_ was evaluated from secondary C^α^-chemical shifts using a random coil reference set for intrinsically disordered proteins [47,48,49].

### 2.12. Wide-Field Fluorescence Microscopy

Fibrils of aSN_1–61_ with heparin (1:1.3 molar ratio) and full-length aSN without heparin were produced as described above. After the fibrillation assay, the samples were centrifuged for 1 h (13,000× *g*, 20 °C) before carefully removing the supernatant and extracting 10 μL of pellet material. The pellet was fully resuspended in 50 μL of primary antibody (mouse IgM anti-heparan sulfate, Cat# 370255-1, Amsbio, Abingdon, UK) at a 1:100 dilution with 1 mg/mL bovine serum albumin (BSA), 100 μM ThT in PBS. After 1 h of incubation at room temperature, the samples were centrifuged (13,000× *g*, 20 °C) for 20 min, and the supernatant was removed before resuspending the pellet in 100 μL of PBS. Following two additional rounds of washing, the pellet was resuspended in 50 μL of Alexa Flour^TM^ 647 labeled secondary antibody (goat anti-mouse IgM, Cat# A21238, ThermoFisher, Denver, CO, USA) at 1:250 with 1 mg/mL BSA in PBS. The secondary antibody was left to react for 30 min at room temperature before the samples were subjected to three additional washing cycles (alternating rounds of 20 min of centrifugation at 13,000× *g* and 20 °C followed by resuspension in PBS). Lastly, the pellets were resuspended in 50 μL of PBS and deposited onto round coverslips pre-coated with a solution of 0.1 g/L poly-L-lysine. The samples were left to disperse for 10 min before tilting the coverslip to remove excess sample and mounting with ProLong Gold Antifade Mountant (ThermoFisher, Denver, CO, USA). The mounted slides were kept in the dark at 6 °C until the time of imaging. Imaging was carried out on a Zeiss AxioImager Z1 Upright Wide-field Microscope (Zeiss, Jena, Germany) using a Plan-Apochromat 63x/1.4 oil-immersion objective equipped with a HXP120 Fluorescence Light Source and a Zeiss Axiocam 506 mono 6Mpx B&W camera (Zeiss, Jena, Germany) for fluorescence detection. A two-channel sequential acquisition of the ThT (fibrils) and Alexa Flour 647 (Heparin) signal was performed as follows: Channel 1 (ThT), 450–490 nm excitation, 500 ms (full-length aSN) or 5000 ms (aSN_1–61_) exposure, and 500–550 emission. Channel 2 (Alexa Flour 647), 625–655 nm excitation, 400 ms exposure, and 665–715 emission. The images were visualized in ImageJ [50], and a colocalization analysis based on Manders’ Colocalization Coefficients [51] was carried out using the JACoP plugin [52]. The M1 (fractional overlap of ThT with heparin) and M2 (fractional overlap of heparin with ThT) coefficients were calculated for 10 pairs of threshold images, and the means ± standard deviations were determined.

## 3. Results

### 3.1. This aSN_1–61_ Maintains Disorder as in Full-Length aSN

To address if removing the NAC- and the C-terminal regions (residues 62–140) from aSN would impact the structural properties of aSN_1–61_, we subjected it to biophysical analyses using SAXS, and NMR and CD spectroscopy. To enhance the expression of aSN_1–61_, this was coupled to a His_6_-SUMO-tag, leaving it in the supernatant of the lysed cells (Appendix A). Subsequently, the tag was fully removed by His-ULP1 protease (Appendix A), leaving pure aSN_1–61_ from a reverse HisTrap column (Appendix A). This was concentrated on an ion-exchange column (Appendix A), leaving no trace of oligomers or higher-order molecular species, as is otherwise often observed for full-length aSN [53,54].

No concentration dependency of aSN_1–61_ was observed in the concentration range used for SAXS data collection (Appendix A), and further analyses were based on the data collected at the highest concentration (573 μM, Figure 1A). The scattering curve and Kratky plot (Figure 1A,B) were consistent with that expected for an IDP. The apparent average *R_g_* of 24 Å and estimated molecular weight (Appendix A) indicated a monomeric state. Using the EOM [39], a pool of random chain structures based on the sequence of aSN_1–61_ was generated and an ensemble fitting of the experimental data was determined (Figure 1A). The selected ensemble indicated a high degree of flexibility in the protein, as the *R_g_* distribution of the selected ensemble (Figure 1C) was almost as broad as the random pool. In addition, the distribution of the selected ensemble was skewed to the right, indicating aSN_1–61_ to be extended compared to the random pool. The disordered state of aSN_1–61_ was further confirmed with far-UV CD (Figure 1D). Thus, based on solution scattering and CD, the isolated N-terminal fragment, overall, retains similar disorder to that of full-length aSN.

We next analyzed if any structural perturbations from the deletion would occur at the residue level using NMR. First, the ^1^H,^15^N-HSQC spectra of aSN and aSN_1–61_ were compared (Figure 1E), revealing that the vast majority of peaks of aSN_1–61_ overlaid those of full-length aSN. The assignment of the aSN_1–61_ resonances revealed that besides effects from the negative charge at the new C-terminal, only small perturbations of the amide shifts were seen relative to full-length aSN (Figure 1F). When comparing the assigned secondary chemical shifts (SCS) of C^α^ for aSN_1–61_ to those for aSN (Figure 1G), the SCSs of aSN were generally only slightly more positive. This may suggest a redistribution of the ensemble towards a less α-helical structure, i.e., a more extended chain, although the small amplitude together with the small perturbations for the amide shifts suggest this to be modest. A slight redistribution may likely be caused by the removal of long-range ionic interactions previously observed between the negatively charged C-terminal and the positively charged N-terminal [55]. Thus, also at the residue level, aSN_1–61_ is as disordered as full-length aSN.

We also measured by NMR the diffusion coefficients of aSN_1–61_ and aSN and calculated their hydrodynamic radius *R_h_* to be 20.1 and 29.1 Å, respectively (Table 1). A recent method published by Nygaard et al. [56] provides information on the extendedness of the chain from the ratio of *R*_g_/*R*_h_. A similar ratio of 1.2 for both aSN_1–61_ and aSN (Table 1) suggested comparable extendedness for the chains.

In conclusion, the N-terminal 61 residues of aSN constitute, in isolation, a fully disordered but slightly extended chain that maintains most of the full-length properties. Likely due to the removal of the acidic C-terminal, the chain becomes, on average, slightly more accessible than the full-length counterpart.

### 3.2. aSN_1–61_ Lacks Intrinsic Fibrillation Propensity

We next explored the fibrillation properties of aSN_1–61_. The aSN_1–61_ lacks the NAC region, which is the known amyloidogenic region of aSN [6,61], and we therefore expected relatively low fibrillation propensity. A ThT assay was conducted under various conditions, including some known to fibrillate aSN, and the secondary structure was monitored by far-UV CD before and after the experiment. Within a timeframe of 300 h, we observed no increase in ThT fluorescence with aSN_1–61_ (Appendix A), even at concentrations where the full-length aSN was shown to fibrillate (see below). This indicated that aSN_1–61_ did not form β-sheet structures capable of interacting with ThT, and CD spectra recorded before and after the ThT assays confirmed that aSN_1–61_ remained disordered (Appendix A). Incubation was continued for 4 weeks without any indications of fibril formation (data not shown). This led to the conclusion that aSN_1–61_ lacked the intrinsic fibrillation propensity of aSN.

### 3.3. Heparin Induces Fibril Formation of aSN_1–61_

To explore the influence of heparin on aSN_1–61_, we repeated the ThT assay and CD measurements as well as conducting negative-stain TEM on aSN_1–61_, in both the absence and presence of different molar ratios of heparin. We observed an increase in ThT fluorescence for aSN_1–61_ incubated with different molar ratios of heparin (200 μM aSN_1–61_, Figure 2A–C, and 100 μM aSN_1–61_, Figure 3A–D,J,K). A clear dependence on the aSN_1–61_ concentration was seen with a delay in the onset of the elongation phase, e.g., changing from 50–100 (Figure 2C) to ~200 h (Figure 3B) and with corresponding changes in the maximum fluorescence values following an expected approximate two-fold decrease. With higher concentration of heparin, we observed more reproducible fibrillation kinetics, although the stochastic nature of the process implicates thorough reproducibility between triplicates. At low molar ratios of aSN_1–61_/heparin, we observed inconsistencies in the kinetic profiles between triplicates (Figure 3B–D). Most of the profiles deviated from the expected sigmoidal shape and instead showed biphasic behavior (Figure 3C,D). With no fluorescence in our controls without aSN_1–61_ (Appendix A) but the sample conditions otherwise being matched, we conclude that heparin does not directly interact with ThT.

The samples were subsequently analyzed for their content of secondary structures using CD (Figure 2D–F and Figure 3E–H,M,N). Indeed, with increasing amounts of heparin, a concomitant change in the secondary structure of aSN_1–61_ occurred, starting from the disordered state (minimum at ~198 nm) and reaching a β-sheet structure (minimum at ~218 nm) at a molar ratio of 1:0.02 (aSN_1–61_/heparin) (Figure 3H). However, at a molar ratio 1:0.15 and lower, the maximum positive ellipticity shifted from ~198 to ~207 nm, with an increase in ellipticity with lower molar ratios (Figure 3F–H). Rather than the formation of β-sheet structures, this indicated the formation of type 1 β-turn structures [62]. The lower the molar ratio, the more pronounced the shift in the maximum positive peak position became, until a molar ratio of 1:0.015, where only one of the samples showed changes in secondary structure (Figure 3K). Even so, at a very low molar ratio of 1:0.007 (aSN_1–61_/heparin), an increase in ThT signal was observed (Figure 3K), but no changes in secondary structure could be detected by CD (Figure 3N).

For selected samples, fibril formation was confirmed with negative-stain TEM (Figure 2G–I and Figure 3I,L). Fibrils were observed in all samples matched by a corresponding decrease in the concentration of aSN and aSN_1–61_ in the supernatants (Appendix A). At an aSN_1–61_/heparin molar ratio of 1:3, amorphous aggregation was observed as well, shown by dark cloudy smudges on the TEM micrographs (Figure 2G). Amorphous aggregates were additionally observed in a few cases at molar ratios of 1:0.3 and 1:0.02 (Figure 3I,L). At a molar ratio of 1:0.15, the morphology of the fibrils appeared straighter than the other fibrils (Figure 2I). The most homogeneously formed long and twisted fibrils with no signs of amorphous aggregation were seen at a ratio of 1:1.7 (Figure 2H).

These observations led to the conclusion that aSN_1–61_ can form ThT-positive fibrils in the presence of heparin at molar ratio 1:0.02 and above. The most homogeneous fibrils form in the presence of relatively large amounts of heparin—at a molar ratio of 1:1.7 aSN_1–61_/heparin. Intriguingly, and to the best of our knowledge, we observed for the first time the formation of type 1 β-turn structures in aSN_1–61_ induced at very low heparin ratios.

### 3.4. Heparin Has Its Own Concentration-Dependent Structure

As we observed that heparin had different effects on the fibrillation of aSN_1–61_ depending on the molar ratio, we additionally investigated the behavior of heparin alone in different buffers using SAXS. The scattering curves showed an increased influence from the so-called structure factor with increasing heparin concentration by a systematic decrease in the scattering intensity in the low s range (below 0.2 nm^−1^) (Appendix A). This indicated repulsive intermolecular interactions, which were evident when reaching 133 μM heparin in 50 mM NaP, and 333 μM in 50 mM NaP, 150 mM NaCl. The repulsive interactions were even more pronounced in a corresponding set of Tris buffers with varying amounts of NaCl. Thus, for the conditions used with an aSN_1–61_/heparin ratio of 1:1.7 and above, the results could therefore be further complicated by heparin–heparin interactions. With the exception of this single condition (1:1.7), the concentrations of heparin used were all below the concentrations where repulsion was observed in the free heparin solutions, although this boundary may change slightly in the mixed samples. Hence, heparin in solution shows distinct intermolecular interactions, which is not surprising given the high density of charges, which must be considered or avoided when studying the influence of heparin on fibrillation.

### 3.5. Heparin Interaction with aSN_1–61_ Is Specific and Mainly Driven by Electrostatics

Heparin is a polar molecule that contains multiple sulfate groups, resulting in a high density of negative charges [18]. It exists in a heterogeneous population, and the MW ranged in this study from 5 to 40 kDa with an average of 15 kDa. Heparin has several different properties that could be responsible for inducing the fibrillation of aSN_1–61_, including the structure of the sugar backbone, the charged sulfate groups, the highly negatively charged surface or a combination of these. To investigate the different features individually, we employed, in combination, fibrillation assays, secondary structure evaluation by CD and negative-stain TEM to compare fibril morphology.

First, to address if the high net charge of heparin induced or influenced the fibrillation of aSN_1–61_, we incubated aSN_1–61_ (200 μM) with and without heparin (333 μM), in 50 mM NaP buffer with 1 M NaCl. The resulting ThT profiles varied among replicates both with and without heparin. An increase in ThT fluorescence was observed in one sample of aSN_1–61_ without heparin (Figure 4A) with a corresponding small change in the secondary structure (Figure 4E), indicating β-turn formation similar to that observed with low amounts of heparin. In the samples of aSN_1–61_ with heparin in 1 M NaCl, changes in ThT fluorescence were also observed. While one sample showed a strong increase in ThT fluorescence (Figure 4B) with accompanying β-sheet formation (Figure 4F), only a small or no increase in ThT (Figure 4B) and no corresponding changes in secondary structure were seen in the others. Thus, the high concentration of NaCl may, to some degree, hinder the heparin-induced fibrillation of aSN_1–61_, potentially by charge screening, and can also occasionally promote β-turn formation, similar to the effect of heparin. The high salt concentration thus influenced the fibrillation but did not on its own induce β-sheet formation and the fibrillation of aSN_1–61_. The presence of salt alone did not influence ThT fluorescence (Appendix A).

Second, to address the effect of the sugar backbone, we used chondroitin sulfate (CS), which has a similar backbone structure to heparin but is far less sulfated [63]. Fibrillation assays with aSN_1–61_ (200 μM) and CS at molar ratios of 1:2.3 and 1:0.7, respectively, were conducted. We immediately observed a low ThT fluorescence signal for aSN_1–61_ with CS but also in controls with CS alone (Figure 4C and Appendix A). The level of fluorescence dropped linearly during the assay, and after 220 h of incubation, a far-UV CD spectrum of aSN_1–61_ with CS showed an unchanged disordered structure (Figure 4G). As CS did not induce fibrillation, the sugar backbone of heparin is unlikely to induce the fibrillation of aSN_1–61_ on its own.

Third, heparin contains, on average, 2.7 sulfate groups per disaccharide [18,21]. To investigate the role of the density of the sulfate groups, aSN_1–61_ (200 μM) was fibrillated in the presence of 160 mM ammonium sulfate, corresponding to the estimated number of charges on heparin based on a Debye–Hückel analysis. No increase in ThT fluorescence was observed (Appendix A). Thus, the sulfate groups alone did not induce the fibrillation of aSN_1–61_.

Finally, β-casein is a highly abundant milk protein, with a high net charge [64]. To address if a high and localized charge density alone would induce fibrillation, aSN_1–61_ (200 μM) was fibrillated in the presence of β-casein (333 μM). As β-casein is also an IDP, it would contribute strongly to the CD signal and we therefore restricted these measurements to a ThT assay. A steady drop in ThT fluorescence was observed (Figure 4D), and β-casein incubated without aSN_1–61_ showed a similar ThT profile (Appendix A). This implied that a high charge density was not enough to induce the fibrillation pathway for aSN_1–61_ and heparin.

In combination, these studies suggest that it is not a single feature of heparin but a combination that induces the fibrillation of aSN_1–61_, pointing towards a highly specific heparin effect. Furthermore, it suggested that the interaction between aSN_1–61_ and heparin is weak and mainly, but not only, driven by electrostatics.

### 3.6. Heparin Binds Weakly But Preferably to aSN_1–61_

To further understand the molecular details of how heparin induces the fibrillation of the low-propensity fibrillating aSN_1–61_, we mapped out interaction sites by titrating heparin into ^15^N-aSN_1–61_ and ^15^N-aSN. Heparin was added in 7-times excess (1:7) to ^15^N-aSN_1–61_, resulting in small CSPs (Figure 5A). The CSPs were most pronounced for residues M5–E13 but were also substantial for residues in an apparent pattern along the chain (Figure 5A). The CSP analysis suggested that the KTK motifs were involved in but not solely responsible for the interaction, as also hydrophobic residues such as Met, Val and Ala were perturbed. As the populations of free and bound states were in fast exchange on the NMR timescale, we could follow the peak position as a function of added heparin and extract a *K_D_* for heparin binding to aSN_1–61_ of 233 ± 40 μM (Figure 5C,D). Furthermore, we measured the diffusion of aSN_1–61_ in the presence of heparin, revealing an increased *R_h_* of 29.6 ± 1.0 Å from the 20.0 ± 0.1 Å of the free aSN_1–61_, supporting the binding of heparin to aSN_1–61_.

The addition of a 10-times molar excess of heparin to full-length aSN resulted in much smaller CSPs than those seen with aSN_1–61_, suggesting a preference for heparin to bind to the truncated aSN_1–61_ (Figure 5B). No CSP saturation was achieved for aSN, and hence, no *K_D_* was determined (Figure 5E).

In summary, heparin binds weakly to aSN_1–61_, mainly within the N-terminal 15 residues, and prefers aSN_1–61_ to full-length aSN, likely because of increased access to the N-terminal residues.

### 3.7. Heparin and aSN_1–61_ Co-Exist in the Fibrils

We next addressed if heparin co-fibrillates with aSN_1–61_ using wide-field fluorescence microscopy. To test for any nonspecific binding, the fibrils of both aSN_1–61_ (formed in the presence of heparin) and aSN (formed in the absence of heparin) were subjected to secondary immunostaining for heparin. The fibrils were visualized by ThT fluorescence, while the presence of heparin was detected with a secondary antibody (2′Ab) conjugated to Alexa Flour^TM^ 647 (Figure 5F). While the aSN fibrils, visible in the ThT channel as large clusters and smaller fibrils, showed no signal in the heparin channel (Figure 5F, bottom), the aSN_1–61_ fibrils induced in the presence of heparin exhibited a distinct signal, confirming co-existence in the fibril. This signal was mostly overlapping with that of ThT (Figure 5F, top). While the large clusters of fibril material (diameter >5 μm) showed a uniform but diffuse signal of co-localized heparin, the smaller clusters (5 μm > diameter > 1 μm) were decorated with a punctate and brighter pattern (Figure 5F,G). This feature was further pronounced in the smallest of the visible assemblies (diameter >350 nm), where bright dots of heparin were found scattered along the axis of the fibrils (Figure 5G). Based on threshold images, the Manders’ Colocalization Coefficients were determined for the fractional overlap of ThT with heparin, M1, and heparin with ThT, M2, (Figure 5H). Here, 37 ± 11% of the ThT signal co-localized with heparin, with the additional ThT signal observed either being a result of heparin being removed in the washing steps or the presence of fibrils consisting only of aSN_1–61_. In the latter case, this could be a result of secondary nucleation and seeding effects from the co-fibrils. The heparin signals co-localized fully with ThT.

Taken together, these results suggest that heparin coexists with aSN_1–61_in the fibrils.

### 3.8. aSN_1–61_/Heparin Co-Fibrils Seed Fibrillation of Full-Length aSN

Having established that the presence of heparin leads to aSN_1–61_ co-fibril formation, we also investigated if the co-fibrils could induce further fibrillation of both full-length aSN and aSN_1–61_. Co-fibrils formed from aSN_1–61_/heparin at a molar ratio of 1:1.7 (aSN_1–61_ 200 μM, heparin 333 μM) were therefore used as seeds.

For aSN_1–61_, controls without seeds followed the previous observations with fibrillation after ~100 h in the presence of heparin and no fibrillation in the absence of heparin (Figure 6A). With aSN_1–61_ in the presence of co-fibril seeds, and with co-fibril seeds and additional free heparin, we observed no lag phase but a continued increase in ThT fluorescence directly from the initiation of the experiment (Figure 6A). Upon seeding, the direct entrance into the elongation phase is consistent with aSN_1–61_ fibrillating into further (co-)fibril material with additional heparin. As this was also observed without additional free heparin (Figure 6A), aSN_1–61_ fibrillation seeded by co-fibril material may also occur without incorporating heparin, consistent with the surplus ThT signal observed in the wide-field fluorescence microscopy. The maximum ThT fluorescence was reached after ~75 and ~20 h, respectively, the increase being fastest with additional heparin present. This difference in growth rate further underlines that heparin facilitates the growth of the fibrils. Finally, the secondary structure of aSN_1–61_ under the four conditions was investigated. At the end-point of the fibrillation assays, the samples with co-fibril seeds and/or heparin all had a β-sheet structure with a minimum at ~220 nm and a maximum at ~198 nm, although some smaller variations in minimum wavelength were visible. aSN_1–61_ without seeds or heparin remained disordered (Figure 6E) as also determined previously in our study.

The kinetic profiles of the unseeded aSN fibrillated alone and with heparin revealed a similar maximum ThT fluorescence (Figure 6B) and a decreased lag phase in the presence of heparin, consistent with the accelerating effect of heparin observed previously [28,29]. When aSN was fibrillated in the presence of aSN_1–61_/heparin co-fibril seeds without and with additional free heparin, we observed a shortening of the lag phase to ~30 h and a three-fold increase in the final ThT value compared to those observed in the un-seeded conditions (Figure 6B), while the presence of additional free heparin did not alter the kinetics in the seeded conditions. The increased ThT signal in the seeded conditions indicates differences in the amount of fibrillated material or in fibril morphology with differences in the binding of ThT compared to that in the unseeded conditions. In all four end-point conditions, a β-sheet structure was confirmed by CD. The β-sheet structure was more pronounced for aSN fibrillated with co-fibril seeds (Figure 6F).

Micrographs were obtained for aSN fibrillated with co-fibrils seeds of aSN_1–61_/heparin (Figure 6C,D) and for aSN fibrillated alone (Figure 6G,H), and different morphologies of fibrils were potentially observed. Alone, aSN fibrils were straighter (Figure 6G) than aSN fibrils induced by co-fibril seeds, which appeared more curved (Figure 6C). Thus, heparin as well as co-fibrils of aSN_1–61_/heparin can accelerate the fibrillation of full-length aSN. Seeding resulted in fibrils that had similar overall morphology to the co-fibrils of aSN_1–61_/heparin (Figure 2H). In addition, the co-fibrils appear to nucleate the further fibrillation of aSN_1–61_.

## 4. Discussion

aSN is a highly disordered protein with transient long-range interactions between the N- and C-terminal tails, likely driven by electrostatics [55,65]. Removing the acidic C-terminal tail and the NAC domain of aSN led to a more accessible N-terminal that remained disordered. Importantly, on its own, the N-terminal domain did not possess intrinsic fibrillation propensity. Once heparin was added, it promoted access to the fibrillation pathways. Heparin engaged residues throughout the N-terminal up to residue 50 but bound predominantly to the 15 most N-terminal residues, with an affinity in the high micromolar range. Only very weak, non-saturable binding of heparin to full-length aSN was observed. As aSN is a dynamic molecule, it is possible that the C-terminal of aSN can compete with heparin, restricting access in full-length aSN, whereas the positive charges of the N-terminal are free to interact in aSN_1–61_.

Despite the low affinity for heparin for full-length aSN, recent work has shown that GAGs such as heparin induce the fibrillation of full-length aSN at specific molar ratios and that the interactions are mainly with the N-terminal of aSN [28,29]. It is also known that heparin can induce the fibrillation of, for example, Tau [66], a protein involved in Alzheimer’s disease; transthyretin [67]; and non-amyloidogenic proteins [68], with different hypotheses for the heparin’s mode of action, one of which is as a scaffold for fibril formation, supporting fibril core structures.

In this study using aSN_1–61_, the affinity for heparin was in a range that allowed for decomposition of features responsible for the observed effects. We have shown that the fibrillation of the N-terminal domain of aSN (aSN_1–61_) was induced by heparin in a concentration-dependent manner. In our SAXS measurements on different heparin solutions, we observed concentration-dependent intermolecular interactions that were additionally dependent on the salt concentration and buffers used. These results underline the importance of accounting for heparin intermolecular interactions or avoiding them as done here, in particular, in the context of fibrillation, where this can add additional factors to an already-complex system, and hence, comparison between studies becomes challenging. In a systematic approach, we tested the effect of individual properties (crowding, charge, sulfation, etc.) and found that it is not a single feature of heparin but a combination of properties that induces fibrillation. Thus, the observed effects are heparin-specific.

Without heparin, we observed no fibril formation of aSN_1–61_ even with extended incubation times, which is consistent with published data showing that fragment 1–60 does not fibrillate [69]. However, a fragment of aSN composed of residues 1–65 was shown to aggregate and could promote the aggregation of full-length aSN [16], although the structural nature of these aggregates has not been fully characterized. In different previous studies, various fragments of aSN have shown different aggregation or fibrillation potential. Almost all fragments lacking the C-terminal acidic tail, but maintaining the NAC domain, show fibrillation potential (1–95 [16], 1–100 [69,70], 1–108 [71], 1–110 [72,73], 1–120 [11,72,73,74], 1–124 [75] and 1–130 [73]) (Figure 7A). However, also, fragments that lack substantial parts of the N-terminal domain, but again have the NAC domain, showed fibrillation propensity (11–140 [76], 21–140 [76], 31–140 [76], 32–140 [74], 41–140 [76] and 58–140 [74]). When the central NAC region was fully missing or if the fragments only carried parts of the NAC region, fibrillation and/or aggregation was/were limited or not observed (1–60 [69,70], 71–140 [76], 61–78 [77], 68–76 [77], 68–78 [77] and 69–79 [77]). These observations suggest the need for residues within the N-terminal for accessing or initiating the fibrillation pathway. A recent study identified a sequence in the N-terminal (37–42), which may have a controlling role in the fibrillation of aSN [78], further highlighting the importance of the N-terminal in relation to fibrillation. Aligning fragments with indications of their published aggregation and fibrillation potential reveals a consistent region that may contain properties required for the fibrillation of aSN, constituted by residues 58–KTKWQVTNV–66 (Figure 7A).

It has been suggested that the fibrillation of aSN is evolutionarily controlled through the maintenance of the imperfect motifs [15], and the identified region contains the imperfect KTKEQV motif, bridging the N-terminal and NAC domain. In this region, βSN and γSN are different to aSN, and at position 63, they have an alanine instead of the high β-sheet-propensity-carrying valine [79]. βSN lacks intrinsic fibrillation potential, and γSN shows only minor intrinsic fibrillation potential compared to aSN [80,81], supporting the observation made here and suggesting the properties enabling fibrillation may locate to an even smaller sequence region. Finally, in the cryo-EM structures that have recently become available, the designated region comprising KTKEQVTNV constitutes central parts or interfaces between fibrils (Figure 7B–E) [82]. Although a few polymorph fibril structures have the interface defined by the NAC region [83], it is becoming increasingly evident that the N-terminal region is also directly involved in the fibril formation.

It is also well known that aSN can interact with membranes through the imperfect motifs, leading to formation of helical structures [84]. This interaction has been hypothesized to play a key role in the native function of aSN [85]. Although not systematically affected or solely responsible, the imperfect motifs are also involved in heparin binding and have a distinct resemblance to other known heparin-binding motifs. We did not observe any helical structure formation in aSN_1–61_ upon heparin binding; instead we observed, for the first time, the formation of type 1 β-turn structures. We speculate that this was observed only in the presence of low ratios of heparin and only in aSN_1–61_, as it is otherwise a transiently formed structure that quickly develops into structures dominated by β-sheets.

Indeed, biphasic behavior was observed for some of the ThT profiles, which could lead to the speculation that heparin needs to interact with aSN_1–61_ to induce structural changes that lead to fibril formation. It has been suggested that heparin does not promote the fibrillation of aSN by interaction but rather stimulates a transition state that favors the β-sheet structure, eventually leading to fibril formation [30]. Based on our results, where we observed a direct interaction with heparin captured more clearly in the truncated aSN variant, we suggest that the fibril formation of aSN_1–61_ stimulated by heparin is based on binding-induced structural changes in the N-terminal, likely through an initial stabilization of type I β-turns. In support of this notion, the β-wrap in AS69 has been shown to stabilize and induce a β-hairpin structure in aSN residues 37–54; a structure that contains a β-turn [86,87] and that is included in our truncated aSN variant. A distinct β-turn structure such as this may be highly relevant for transit into the fibrillation pathway and the further formation of β-sheets.

Heparin may also function as a scaffold and/or structural support for fibril formation. In the case of Tau, the presence of heparin induces different structural polymorphic forms, and in the corresponding cryo-EM structures, density can be observed in proximity of, for example, the outer lysines, potential interaction sites of heparin [88]. In a study of heparin-induced β-endorphin fibrillation, the co-existence of heparin with the fibrils, and thus its existence as a direct structural component, was confirmed using two-color direct stochastic optical reconstruction microscopy [89]. For both Tau and aSN, internalization and prion-like propagation are mediated through binding to cell-surface HSPGs [22], further emphasizing the importance of understanding the interactions with GAGs in different states populated during the fibrillation process. Using fluorescence microscopy, we observed a clear co-localization of heparin and the fibril, and in the wide-field microscopy and seeding experiment, we observed the potential for the secondary nucleation and formation of fibrils without heparin as part of the structure. Although we determined a low affinity for the aSN_1–61_ monomer, and an even lower one for aSN, it is likely to be enhanced in the oligomer and the fibrils through an avidity or local concentration effects, but given the need for heparin to induce the fibrils, we were not able to decipher these affinities.

A key observation made in this work was that heparin at low concentrations bound to the N-terminal of aSN and induced type-I β-turn structures that are likely a key passageway for entering the fibrillation pathway. Thus, heparan sulfates of the cell surface HSPGs may not only internalize the fibrillar aggregates but also stabilize and/or induce the initial fibrillation-prone states [90]. If this is indeed so, and this fibrillation triggering also happens in vivo, then the type-I β-turn structures constitute a scaffold for potential targeting by small molecule drugs to be further used therapeutically. To address this further, more structural information including the stabilization of the particular structure is needed, and the use of a smaller fragment of the N-terminal would be a relevant starting point.

Access to the N-terminal is easier and less sterically hindered in the truncated aSN than that in the full-length counterpart, and although the affinity of heparin for aSN was low, heparin still also promoted the fibrillation of full-length aSN. In a cellular context, increased access to the N-terminal may occur in the Ca^2+^-bound state [91], and increased serum Ca^2+^-levels have been reported in patients with Alzheimer’s disease with Lewy Bodies [92]. Similarly, in the growing fibril, the local concentration of the N-terminal would be higher and, in the inclusions, HPSG and aSN would be concentrated, increasing the in vivo relevance of our studies. Conversely, membrane binding as well as the phosphorylation of aSN may lead to restricted access to the N-terminal. How these changes in context affect heparin binding remains to be addressed. For an IDP such as aSN, multifunctionality is the key, and thus, the overall functional outcome depends critically on the distribution of states as well as on their availability [93,94]. Thus, tipping the balance in the populations of aSN by Ca^2+^ or membrane binding, or by phosphorylation or mutations, may not change the size of each population dramatically but can have a large functional impact, including under pathophysiological conditions. Decomposing all the relevant states of aSN is therefore important for understanding its function as an IDP. The heparin bound form is one.

## 5. Conclusions

By removing the complexity arising from the process of aSN fibrillation by focusing on the aSN_1–61_ variant, we observed direct binding of heparin to the N-terminal ~50 residues of aSN. The aSN_1–61_ variant did not possess intrinsic fibrillation potential, but heparin could act as a promoter for initiating fibrillation, via the induction of type-I β-turn structures in the aSN N-terminal. These structures were observed by CD spectroscopy through a distinct ellipticity profile. Heparin co-existed with aSN_1–61_ in the formed fibrils, likely in the form of a structural scaffold, and the co-fibrils seeded additional fibrillation even in the absence of free heparin, as well as of full-length aSN, resulting in fibrils with a potentially different morphology compared to that of fibrils formed by aSN alone. Thus, small amounts of heparin, or more likely that in the form of aged heparan sulfate with an increased level of sulfation, present in Lewy bodies of PD patients, can potentially initiate a cascade of fibrillation-relevant events.

## Figures and Tables

**Figure 1 biomolecules-10-01192-f001:**
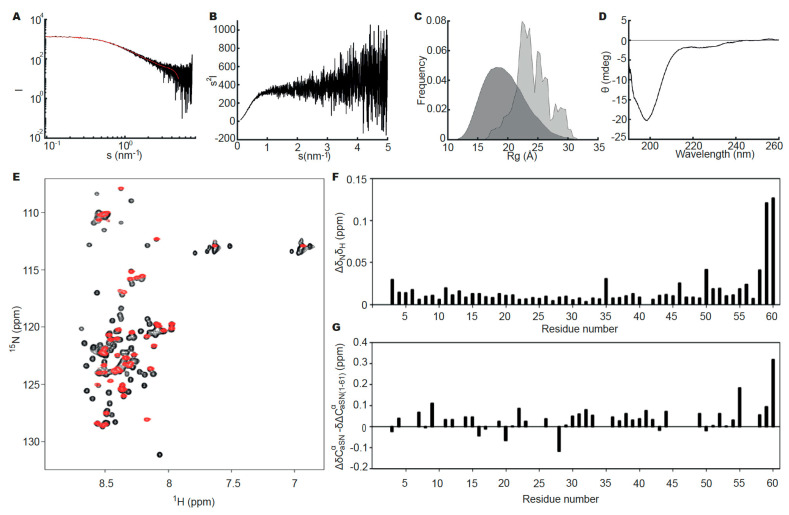
Structural studies of aSN_1–61_. (**A**) Experimental small angle X-ray scattering (SAXS) data, relative scale, for aSN_1–61_ (573 μM; black) and fit (red) obtained using ensemble optimization method (EOM) (χ^2^ = 0.59). (**B**) Kratky plot of the SAXS data for aSN_1–61_. (**C**) *R*_g_ distribution of the random pool (dark grey) and the selected ensemble (light grey) from EOM. (**D**) Far-UV CD spectrum for 10 μM aSN_1–61_. (**E**) ^1^H,^15^N-HSQC spectra for ^15^N-aSN (black) and ^15^N-aSN_1–61_ (red). (**F**) Amide chemical shift perturbations between aSN and aSN_1–61_. (**G**) Differences in secondary chemical shifts of C^α,^ between aSN and aSN_1–61_. Data from the C-terminal residues have been omitted for clarity.

**Figure 2 biomolecules-10-01192-f002:**
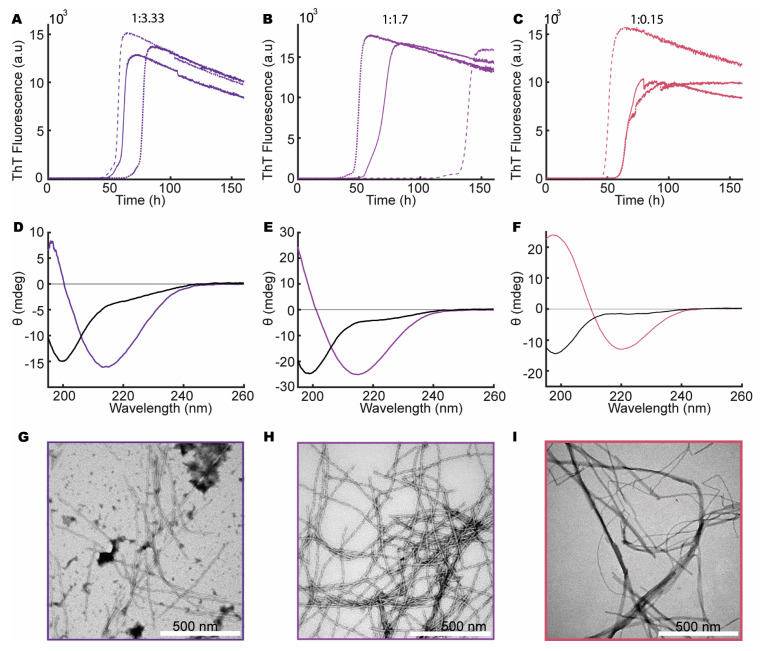
aSN_1–61_ fibrillation in the presence of heparin with a fixed aSN_1–61_ concentration of 200 μM. Molar ratios of aSN_1–61_/heparin are 1:3.3 (dark purple), 1:1.7 (light purple) and 1:0.15 (red). (**A**–**C**) ThT assays in triplicate at the different molar ratios as indicated on top. (**D**–**F**) Far-UV CD spectra of aSN_1–61_/heparin at different molar ratios before (black) and after fibrillation (color corresponding to ThT curves). (**G**–**I**) TEM micrographs of co-fibrils at the corresponding molar ratios of aSN_1–61_/heparin.

**Figure 3 biomolecules-10-01192-f003:**
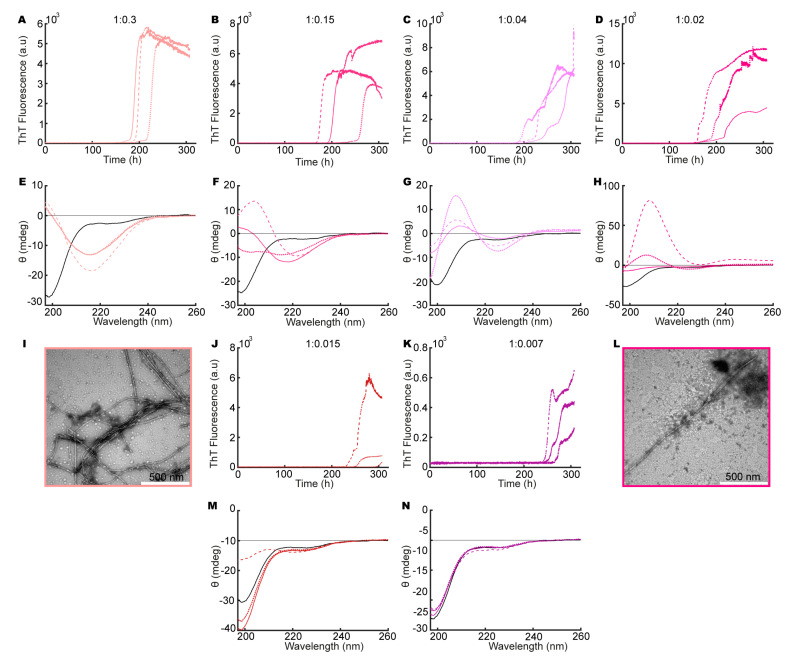
aSN_1–61_ fibrillation in the presence of heparin with a fixed aSN_1–61_ concentration of 100 μM. Molar ratios of aSN_1–61_/heparin are (**A**,**E**,**I**) 1:0.3 (salmon), (**B**,**F**) 1:0.15 (dark pink), (**C**,**G**) 1:0.04 (light purple), (**D**,**H**,**L**) 1:0.02 (pink), (**J**,**M**) 1:0.015 (dark red) and (**K**,**N**) 1:0.007 (purple). (**A**–**D**,**J**,**K**) ThT assays in triplicate at different molar ratios of aSN_1–61_/heparin. Triplicates are shown by different line types. (**E**–**H**,**M**,**N**) Far-UV CD spectra at different molar ratios of aSN_1–61_/heparin, before (black) and after (color and line types corresponding to ThT curves) fibrillation. (**I**,**L**) TEM micrographs of co-fibrils at selected molar ratios of aSN_1–61_/heparin.

**Figure 4 biomolecules-10-01192-f004:**
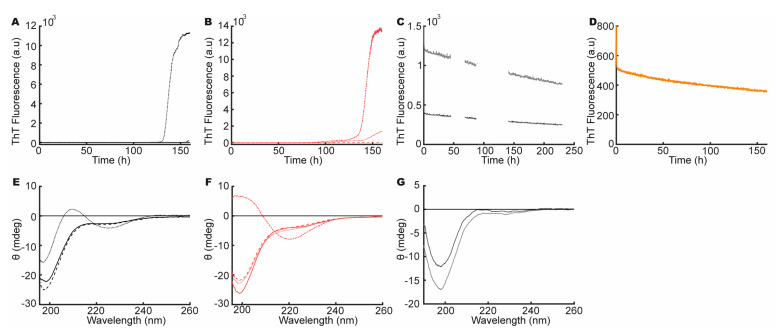
Features of heparin influencing aSN_1–61_ fibrillation. (**A**) ThT assay of aSN_1–61_ alone (black) in 1 M NaCl and (**B**) at a molar ratio 1:1.7 of aSN_1–61_/heparin (red) in 1 M NaCl. (**C**) ThT assay of aSN_1–61_ incubated with CS in estimated molar ratios of 1:2.3 (light grey) and 1:0.7 (dark grey) The missing values in ThT fluorescence around 50 and 100 h are due to a machine error. (**D**) ThT assay of aSN_1–61_ incubated with β-casein in molar ratio 1:1.7 (orange). (**E**) Far-UV CD spectrum of aSN_1–61_ (black) in 1 M NaCl before (solid) and after (dotted and dashed) fibrillation. (**F**) Far-UV CD spectrum of aSN_1–61_:heparin (red) in 1 M NaCl before (solid) and after (dotted and dashed) fibrillation. (**G**) Far-UV CD spectrum after fibrillation of aSN_1–61_ with chondroitin sulfate (CS) in molar ratios of 1:2.3 (light grey) and 1:0.7 (dark grey). No CD data are reported with β-casein as this would dominate the spectrum.

**Figure 5 biomolecules-10-01192-f005:**
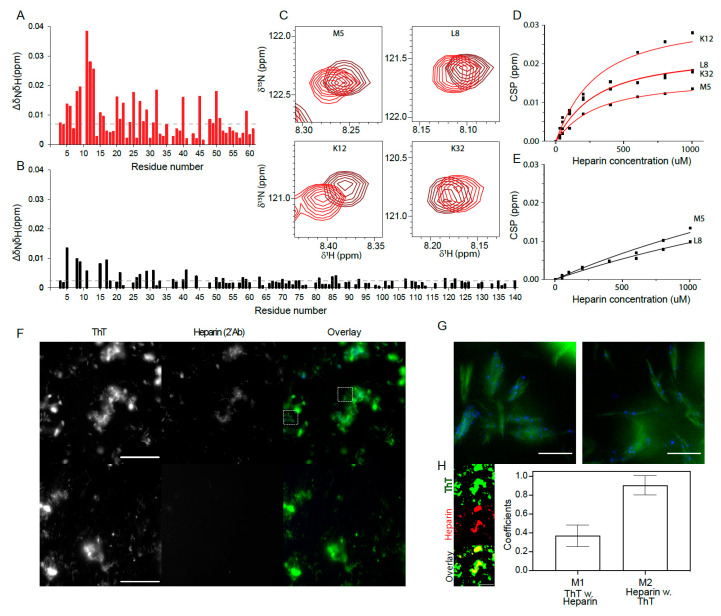
aSN and heparin interaction. (**A**) Amide chemical shift perturbations (CSPs) of aSN_1–61_ upon addition of heparin (1:7); the black line represents the average CSPs. (**B**) Amide CSPs of aSN upon addition of heparin (1:10); the red line represents the average CSPs. (**C**) Zoom on the peak positions of specific residues of aSN_1–61_ in the absence (red) and presence (brown) of heparin. (**D**) CSPs of the most perturbed residues of aSN_1–61_ as a function of added heparin. The solid line represents a fit to a one-site binding model (cf. experimental procedures). (**E**) CSPs of aSN upon addition of heparin as a function of added heparin. The solid line represents a fit to a one-site binding model as described in [46]. (**F**) Wide-field fluorescence microscopy images of (top) aSN_1–61_ fibrils produced in the presence of heparin (1:1.3 molar ratio) and (bottom) aSN fibrils produced in the absence of heparin. The (left) ThT signal from the fibrils is shown alongside the (middle) Alexa Flour 647 signal from immunostained heparin and the (right) overlay with highlighted regions of interest (ROIs). Scale bars are 50 μm. (**G**) Zoom of the ROIs from the overlay in (**F**). Scale bars are 5 μm. (**H**) Example threshold images of ThT (green, top), heparin (red, middle) and overlay (bottom) used to calculate the depicted Manders’ Colocalization Coefficients (means ± s.d., n = 10) of M1 (ThT with heparin) and M2 (heparin with ThT). Scale bar in the threshold image examples are 50 μm.

**Figure 6 biomolecules-10-01192-f006:**
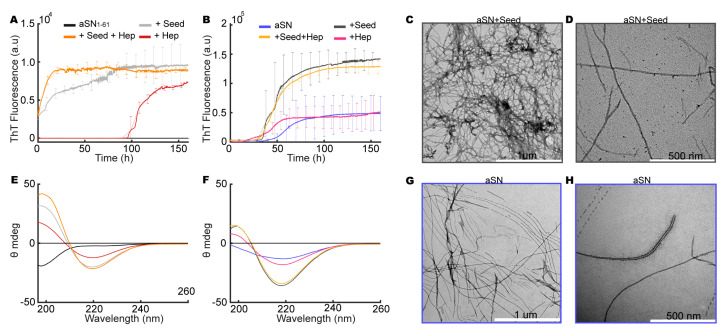
Seeding with aSN_1–61_/heparin co-fibrils. (**A**) ThT assays for aSN_1–61_ (black), aSN_1–61_ with heparin (red), aSN_1–61_ with co-fibril seeds (grey), and aSN_1–61_ with co-fibril seeds and heparin (orange). (**B**) ThT assay for aSN (blue), aSN with heparin (magenta), aSN with co-fibril seeds (dark grey), and aSN with co-fibril seeds and heparin (yellow). Note that the *y*-axis ranges depicted differ for the two fibrillation assays (**A**,**B**). (**C**) Far-UV CD spectra of aSN_1–61_, aSN_1–61_ with heparin, aSN_1–61_ with co-fibril seeds, and aSN_1–61_ with co-fibril seeds and heparin (colors as in (**A**)). (**D**) Far-UV CD spectra of aSN, aSN with heparin, aSN with co-fibril seeds, and aSN with co-fibril seeds and heparin (colors as in (**B**)). (**E**,**F**) TEM micrographs of aSN fibrillated with seeds. (**G**,**H**) TEM micrographs of fibrillated aSN.

**Figure 7 biomolecules-10-01192-f007:**
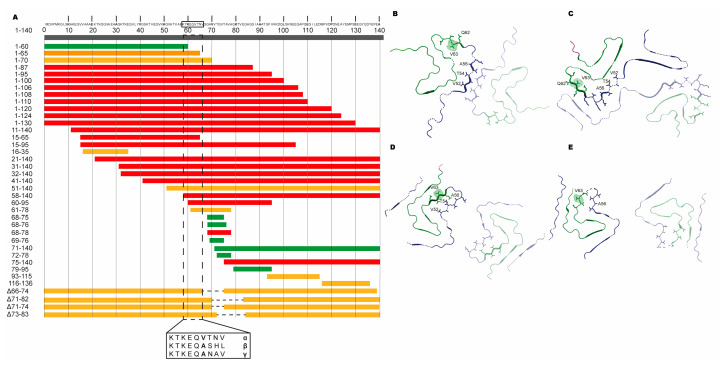
Key fibrillation-promoting regions of aSN. (**A**) The red bars indicate that fibrils have been reported in the literature. The yellow bars indicate that aggregates or not-further-characterized fibril material has been reported in the literature. The green bars indicate that neither fibrils nor aggregates have been observed. The position of residue 58–66 is outlined with a box where the corresponding sequences of α-, β- and γ-SN are shown. In Appendix A, all the fragments are provided along with the references. (**B**–**E**) Pairs of aSN molecules extracted from cryo-EM structures of aSN fibrils where the N-terminals are shown in blue and NAC regions, in green. Residues 52 to 63 are shown as sticks, and residue 63 is additionally highlighted in spheres. The corresponding PDB codes are (**B**) 6A6B, (**C**) 6XYO, (**D**) 6RT0 and (**E**) 6RTB.

**Table 1 biomolecules-10-01192-t001:** Hydrodynamic radius and radius of gyration of aSN and aSN_1–61_.

		Radii (Å)		
		aSN	aSN_1–61_	Reference
R_g_	Folded	14.4	10.5	[57]
	Unfolded	36.8	22.4	[58]
	SAXS	36.5	24.0 ± 0.3	[59], This work
R_h_	Folded	20.1	15.8	[45]
	Unfolded	36.8	22.9	[45]
	IDP	31.1	20.3	[60]
	NMR *	29.1 ± 0.1	20.1 ± 0.1	This work
R_g_/R_h_	Folded	0.71	0.66	[57]
	Unfolded	1	0.97	[58]
	Nygaard	0.98	1.05	[56]
	SAXS/NMR	1.08	1.2	This work

*: Errors from fit.

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
