# Peer review of "The Non-Fibrillating N-Terminal of α-Synuclein Binds and Co-Fibrillates with Heparin"

_biomolecules, 2020, doi:10.3390/biom10081192_

Round 1

Reviewer 1 Report

The manuscript of ”The Non-Fibrillating N-terminal of a-Synuclein  Binds and Co-Fibrillates with Heparin” by Line K. Skaanning, Angelo Santoro, Thomas Skamris, Jacob Hertz Martinsen, Anna Maria D’Ursi, Saskia Bucciarelli, Bente Vestergaard, Katrine Bugge, Annette E. Langkilde and Birthe B. Kragelund addresses very important issue of the mechanisms of fibrillation of alpha-synuclein and how it is affected by heparin, which is relevant to the Lewy body formation in Parkinson’s disease. The authors have used a wide range of biophysical techniques such as tioflavin-T fluorescence assay, circular dichroism, NMR and NMR diffusion experiments, transmission electron microscopy and wide field fluorescence microscopy to characterise from different angles the fibrillation and interaction with heparin of N-terminal 61 amino acid residues sequence of alpha-synuclein, which does not self-assembly into the amyloid structures in the absence of heparin. The authors have revealed that heparin indeed interacts and promotes fibrillation of the N-terminal part of alpha-synuclein via inducing the beta-turn in its natively unfolded structure and remains as a part of the amyloid scaffold after fibrillation process is completed. The research is very detailed, technically and scientifically sound and clearly presented. I recommend this manuscript to the Biomolecules without reservation and believe that it will be of great interest to a broad scientific community and have significant impact on the understanding the mechanisms leading to Parkinson’s pathology.

Author Response

We thank the reviewer for the recommendation and for the positive comments about our work and its significance to the field.

Reviewer 2 Report

The authors present a well-executed biophysical and biochemical study on the effects of heparin, a highly sulphated glycosaminoglycan, on the fibrillization of a truncated fragment of the aSN protein, which includes the whole of the N-terminal region (1-61).

It is already well known that the negatively-charged heparin polymer accelerates fibrillation of whole-length aSN, potentially through ionic interactions with the N-terminal end, which is positively-charged. Hence, the present manuscript can be considered as an incremental study with regards to impact and significance in the field.

The authors start by demonstrating that the aSN(1-61) fragment is in a monomeric and intrinsically disordered state like the full-length protein. The aS(1-61) peptide does not aggregate into fibrils, even after 4 weeks of incubation. However, using a powerful combination of ThT fluorescence assays, far-UV CD spec measurements and TEM imaging, they show that in the presence of heparin, structural changes of the aSN(1-61) peptide occur. These include formation of an interesting transient type 1 beta-turn structure at low heparin concentrations, to a characteristic beta-sheet structure with increasing amounts of heparin. The far-UV ellipticity profile demonstrating the former is convincing. Using appropriate compounds and controls, they also conclude that the interaction between heparin and aSN(1-61) is (predictably) mainly driven by electrostatic interactions with the KTK-motifs in the first 15 N-terminal residues – while the sole presence of sulfate groups, a high charge density or a sugar backbone alone did not suffice to induce fibrillization of aSN(1-61). KD for heparin binding to aSN(1-61) was quite high (~200-280 μM). Finally, the authors used fluorescence microscopy images to confirm the co-localization of heparin in the aSN(1-61) fibrils. The co-fibrils of aSN(1-61) and heparin accelerated fibrillization of both aSN(1-61) and full-length aSN.

Conducted experiments were technically sound, including appropriate controls and adequate replication. Perhaps one could comment on the variability in the ThT kinetic profiles of aSN(1-61) fibrillization at low heparin concentrations. However, this is understandable given the stochastic nature of the process and the low heparin affinity, and does not compromise the authors’ conclusions.

The interpretation of the data is proper and supports the reported findings.

On the other hand, the significance of these results is not so clear and should be better addressed in a revision. For instance:

  1. The rationale behind the study is that heparin sulfate proteoglycans have been shown to be components of Lewy bodies from PD patients. However, the aSN deposits in Lewy body diseases are intracellular, and thus it is less clear how glycosaminoglycans may be involved in initiating aSN fibrillization in vivo.
  2. Given the extraordinarily high (typically ~200-300 μM) concentrations of aSN peptides and heparin used in these experiments, coupled with the fact that the affinity of heparin for aSN is pretty low, how confident are the authors that their in vitro studies accurately model in vivo interactions?

Minor Comments

  1. Revise Abstract to improve flow and English, e.g. lines 24-26 “Via isolation….be decomposed”
  2. Line 12 – replace ‘2’ by ‘3’
  3. Line 95 – delete extra letter ‘M’

Author Response

We thank the reviewer for the positive commenting on our work and have detailed the answers as well as the correction made below for each point raised:

Conducted experiments were technically sound, including appropriate controls and adequate replication. Perhaps one could comment on the variability in the ThT kinetic profiles of aSN(1-61) fibrillization at low heparin concentrations. However, this is understandable given the stochastic nature of the process and the low heparin affinity, and does not compromise the authors’ conclusions.

The stochastic nature of the process indeed introduces variability in the kinetic profiles. As suggested by the reviewer, we have inserted a comment on this in the main manuscript (l. 360). Thank you for pointing this out.

The interpretation of the data is proper and supports the reported findings.

On the other hand, the significance of these results is not so clear and should be better addressed in a revision. For instance:

1. The rationale behind the study is that heparin sulfate proteoglycans have been shown to be components of Lewy bodies from PD patients. However, the aSN deposits in Lewy body diseases are intracellular, and thus it is less clear how glycosaminoglycans may be involved in initiating aSN fibrillization in vivo.

In addition to the GAGs found in the Lewy bodies, heparan sulfate proteoglycans have been shown to also play a part in the internalization and spreading of the pathogenic forms, through fibril and/or aggregate binding to the heparan sulfate on the cell surface or specific glycosylated surface proteins. Heparan sulfate and heparin clearly induce fibrillation in vitro, yet whether this induction also happens in vivo, on the cell surface, remains elusive. This may not have been too clearly highlighted neither in the introduction or the discussion and we appreciate the comment to increase this focus. We have addressed this in the introduction (l. 65-67) and in the discussion (l. 693-695, 705-707)

2. Given the extraordinarily high (typically ~200-300 μM) concentrations of aSN peptides and heparin used in these experiments, coupled with the fact that the affinity of heparin for aSN is pretty low, how confident are the authors that their in vitro studies accurately model in vivo interactions?

This is a relevant point to raise. The N-terminal alone does not exist in vivo, but studied in isolation have allowed access to a more readily studied interaction with heparin which also occurs in the full-length protein. Thus, the amplification from studying the N-terminal alone, as well as using the high protein concentrations, was needed. As we have speculated on in the discussion, accessibility to the N-terminal of aSN may be increased by Ca2+-binding, as well as in the forming fibrils, in which the local concentration would be even higher than 200 -300 μM. Furthermore, the concentration ratio of GAGs to aSN would be assumed to be higher in the inclusions. We have expanded our considerations of these issues in the discussion (l. 716-718) and thank the reviewer for raising it.

Minor Comments

1. Revise Abstract to improve flow and English, e.g. lines 24-26 “Via isolation….be decomposed”

This has been reformulated in the revised version. The original sentence did not make sense. Thanks

2. Line 12 – replace ‘2’ by ‘3’

Done

3. Line 95 – delete extra letter ‘M’

Done